# Investigation and Determination of Kinetic Parameters of Sweeteners Based on Steviol Glycosides by Isoconversional Methods

**DOI:** 10.3390/foods14071233

**Published:** 2025-03-31

**Authors:** Naienne da Silva Santana, Sergio Neves Monteiro, Tatiana Carestiato da Silva, Michelle Gonçalves Mothé

**Affiliations:** 1Department of Organic Processes, School of Chemistry, Federal University of Rio de Janeiro, Rio de Janeiro 21941909, Brazil; naienne.santana@gmail.com; 2Department of Science and Technology, Military Engineering Institute, Rio de Janeiro 22290270, Brazil; snevesmonteiro@gmail.com; 3National Institute of Industrial Property, Rio de Janeiro 20090910, Brazil; tatiana_carestiato@hotmail.com

**Keywords:** sweeteners, steviol glycosides, erythritol, xylitol, decomposition, activation energy

## Abstract

In this study, the decomposition processes of tabletop sweeteners based on steviol glycosides were investigated to determine the kinetic parameters of activation energy (E_a_) and the logarithm of the pre-exponential factor (ln A) based on the converted fraction (α). These parameters were assessed using the Friedman and Ozawa–Flynn–Wall isoconversion methods with the NETZSCH Kinetics Neo software and the Model Free package. This study also aimed to explore the probable mechanism of the thermal decomposition of these materials. The thermal degradation of the samples was carried out in a temperature range of 150 to 400 °C under nitrogen flow, with heating rates of 5, 10, and 20 °C min^−1^. The results indicated that both stevioside and steviol glycoside (E3) samples require higher energy to initiate their decomposition. Furthermore, the samples based on steviol glycosides exhibited distinct probable decomposition mechanisms: a model of two consecutive reactions followed by two competitive reactions for stevioside (FnFnFnFn model), three consecutive stages for the steviol glycoside sample (FnCnFn model), two consecutive stages for the steviol glycoside + erythritol sample (FnCn model), and three consecutive stages for the steviol glycoside + xylitol sample (FnFnFn model).

## 1. Introduction

Since chronic diseases such as obesity and diabetes have become a public health issue, consumers have been seeking healthier, lower-calorie diets featuring natural ingredients. This growing health awareness has driven the demand for natural, non-caloric sweeteners such as stevia as an alternative to traditional table sugar. According to IMARC, the global stevia market was valued at approximately USD 818 million in 2024 and is expected to reach over USD 1 billion by 2033 [1].

Steviol glycosides, commonly referred to as stevia, are natural sweeteners extracted from the leaf of the *Stevia rebaudiana* plant. With a high sweetening power (150 to 450 times greater than sucrose) and zero caloric content, these sweeteners have been used as sugar substitutes in various food products and marketed as tabletop sweeteners [2,3]. As tabletop sweeteners, steviol glycosides are generally combined with other sweeteners to improve their flavor profile and, consequently, their acceptance by consumers [4,5]. When used to prepare food or beverages, temperature and time can lead to sweetener degradation, forming other compounds that may present undesirable sweetening properties and pose health risks. Therefore, the present study investigates the thermal decomposition kinetics of stevia-based tabletop sweeteners to propose potential reaction mechanisms.

Determining kinetic parameters such as activation energy (E_a_) and the pre-exponential factor (ln A) is essential for analyzing reaction mechanisms. Thermal analysis techniques effectively evaluate the physical changes or chemical reactions that materials undergo under various conditions. There are several approaches in the literature for determining kinetic parameters for different materials, including model-free and model-based methods. The International Conference on Thermal Analysis and Calorimetry (ICTAC) advocates prioritizing model-free methods, particularly complex reaction process mechanisms, as they are generally more reliable [6,7].

Among the model-free methods, the Friedman and Ozawa–Flynn–Wall isoconversional methods are commonly used to determine E_a_ and ln A. These methods do not require prior knowledge of the reaction mechanism and can evaluate kinetic parameters for each degree of conversion (α) as the reaction progresses [6,7].

Additionally, several classical kinetic models documented in the literature have been studied and adapted to assess decomposition processes by thermal analysis. Notable examples include the methods of Newkirk [8], Coats and Redfern [9], Doyle [10], and Ingraham and Marier [11].

Few studies have examined the thermal decomposition kinetics of sweeteners, with most focusing on compounds of artificial origin [12,13,14]. One study evaluated the decomposition kinetics of saccharin using the Ozawa–Flynn–Wall method, reporting an average E_a_ value of 80 kJ mol^−1^ for the compound [13]. In contrast, another study investigated the isothermal kinetics of a standard sample of aspartame and an aspartame-based tabletop sweetener, observing that the commercial tabletop sample presented the lowest activation energy value, around 118 kJ mol^−1^ [12]. To continue the previous thermal studies of commercial samples based on stevia and given the complexity of their thermal decomposition profiles [15,16], the present study aimed to determine the kinetic parameters of six different samples of natural-origin tabletop sweeteners based on steviol glycosides using Kinetics Neo software version 3.0.1, applying both the isoconversional methods.

## 2. Materials and Methods

### 2.1. Materials

Commercial samples of natural tabletop sweeteners in granular form were analyzed. The composition listed on their packaging included the following ingredients: stevioside (Sigma-Aldrich), E2 (steviol and erythritol glycosides), E3 (steviol glycosides), E4 (steviol and xylitol glycosides), erythritol, and xylitol. However, the packaging labels did not disclose the quantity or percentage of each ingredient. Table 1 shows some characteristics of the natural sweeteners.

### 2.2. Kinetic Study

This paper investigates the kinetic parameters using the model-free approach, which includes differential (Friedman) and integral (Ozawa-Flynn-Wall) isoconversional methods. The selection of these methods is based on the advantage that isoconversional methods do not require any assumptions of the reaction models. As a result, the activation energy can be determined without pre-defining the reaction function. Exploring complex solid-state thermal decomposition processes also necessitates a wide range of experimental conversions at specified temperatures.

As described in Equation (1), the thermal degradation rate of a specific sample can be characterized in terms of the extent of conversion (*α*) and temperature (*T*).(1)$$\frac{d\alpha }{dt}=-k\left(T\right)\;.\;f(\alpha )$$
where;

*α =* decomposed fraction; *t* = time; *k* = rate constant; *T* = temperature in Kelvin; *f(α)* = kinetic model.

The degraded mass portion of the sample is represented in Equation (2):(2)$$\alpha =\frac{m_{i}-m_{t}}{m_{i}-m_{f}}$$
where;

*m_i_* = initial mass; *m_t_* = time-specific mass; and *m_f_ =* final mass.

Furthermore, the Arrhenius equation (Equation (3)) defines the temperature-dependent function in terms of activation energy and pre-exponential factor:(3)$$k\left(T\right)=\;A.\;{exp}^{-E_{a}/RT}$$
where;

*A =* pre-exponential factor; *E_a_* = activation energy; *R* = Universal Gas Constant.

The samples are subjected to various linear heating rates, denoted as *β* (Equation (4)).(4)$$\beta =\frac{dT}{dt}$$

By combining Equations (1), (3), and (4), the rate equation can be expressed as Equation (5):(5)$$\frac{d\alpha }{dT}=\frac{k_{0}}{\beta }\;.{exp}^{(-E_{a}/RT)}\;.\;f(\alpha )$$

#### 2.2.1. Friedman Method

Substituting the heating rate equation *β* (Equation (4)) into Equation (5) and then applying the natural logarithm to both sides results in the Friedman method equation (Equation (6)). The parameters *E_a_* and *A* are calculated for the conversion (*α*) at different stages with the assistance of software.(6)$$ln\left(\frac{d\alpha }{dt}\right)=-\frac{E_{a}}{RT}+ln\left[A\;.\;f(\alpha )\right]$$
where;

*α =* conversion reaction degree; *t =* time; *E_a_* = activation energy; *T* = temperature in Kelvin; *R* = Universal Gas Constant; *A* = pre-exponential factor, *f(α) =* conversion function of *α* expressing the dependence of the physical properties.

The Friedman method [23] enables the determination of the activation energy without making assumptions about the reaction type. It is accomplished using data points with the exact conversion obtained from measurements with different heating rates or under isothermal conditions.

#### 2.2.2. Ozawa–Flynn–Wall Method (OFW)

The integral isoconversional analysis method calculates the changes in apparent activation energy in thermogravimetric curves obtained at a minimum of three different heating rates. This methodology is quantitatively suitable for multistep processes.

The activation energy (*E_a_*) can be determined by plotting the logarithm of the heating rates (log *β*) versus *1000/T*, resulting in a linear relationship for each conversion degree (*α*), as shown in Equations (7)–(12) [24,25,26,27].

Basic expressions: *α* = *φ*(*θ*)(7)$$\theta =\int_{0}^{t}e^{-\frac{E_{a}}{RT}}.dt$$(8)$$\theta =\frac{E_{a}}{\beta R}.p\left(\frac{E_{a}}{RT}\right)$$

*p* = function proposed by Doyle (Equation (9)) [27].(9)$$\log p\left(\frac{E_{a}}{RT}\right)=-2.315-0.4567\left(\frac{E_{a}}{RT}\right)$$

*θ* is constant for a given conversion (Equations (10)–(12)):(10)$$\log\beta +0.4567\left(\frac{E_{a}}{RT}\right)=constant$$(11)$$\frac{\partial log\beta }{\partial \left(\frac{1}{T}\right)}≅-\left(\frac{0.457}{R}\right).E_{a}$$(12)$$E_{a}≅-18.2\left(\frac{\partial log\beta }{\partial \left(\frac{1}{T}\right)}\right)$$

To evaluate the decomposition kinetic parameters of the sweetener samples, thermogravimetric (TG) curves were obtained at three heating rates of 5, 10, and 20 °C min^−1^. A model SDT Q600, TA Instruments (New Castle, DE, USA) simultaneous analyzer was employed, with 10 mg of sample in a platinum crucible within a temperature range of 25 to 600 °C under a nitrogen atmosphere (flow rate of 100 mL min^−1^). Using Kinetics Neo version 3.0.1, Netzsch, the Model Free package was applied to determine the activation energy and pre-exponential factor by the Friedman and Ozawa–Flynn–Wall isoconversional models. In addition, the Model-Based package was utilized to identify the most appropriate reaction order for the thermal decomposition process of the samples. It is worth mentioning that the data generated were obtained through interactions in the software.

## 3. Results and Discussion

The kinetic study was conducted to advance the investigation into the thermal manipulation process of natural tabletop sweeteners subjected to temperature variations [11,12]. This study examined the temperature ranges in which the main degradations of the six samples occurred, specifically between 150 and 400 °C.

The Friedman analyses (Figure 1a and Figure 2a) of the stevioside (standard) and E3 (steviol glycosides) samples displayed a central decomposition peak between 250 and 350 °C. Similarly, the Ozawa–Flynn–Wall analyses (Figure 1b and Figure 2b) showed isoconversional lines with greater parallelism between 270 and 350 °C, corresponding to the region of highest decomposition of these samples. It indicates that model-free analysis explains the decomposition processes of study samples. These similarities suggest that stevioside is the predominant glycoside in the E3 sample. It is important to note that stevioside and rebaudioside A are typically found in more significant proportions in steviol glycoside extracts. However, the industry is working to produce sweeteners with a higher content of other glycosides and a more pleasant flavor to enhance consumer accessibility [28,29,30].

A comparison of the activation energy (E_a_) values of these two samples is shown in Table 2, revealing that the standard stevioside sample requires a higher amount of energy to initiate its concentration process than sample E3. It is also worth noting that the pre-exponential factor (ln A) increases with increasing activation energy, which indicates an increase in collisions between molecules with increasing temperature. The stevioside sample presents higher ln A values than sample E3, suggesting it began its occurrence more quickly.

Sample E2 (steviol glycosides + erythritol) exhibited a peak around 235 and 320 °C in its Friedman analysis (Figure 3a), similar to the erythritol sample (Figure 4a). Both samples also showed greater parallelism of the isoconversional lines between 190 and 305 °C, which include their maximum decompositions (Figure 3b and Figure 4b). Despite being marketed based on stevia, sample E2 presents activation energy values much lower than the stevioside and E3 samples, as shown in Table 3. However, sample E2 presented values that were more similar to those of the erythritol sample. Note that a pure erythritol sample (C_4_H_10_O_4_) needs more energy to initiate the liquidation process and exhibits increasing activation energy values with increasing conversion (*α*).

The E4 sample (steviol glycosides + xylitol) exhibited a peak in the Friedman analysis (Figure 5a) in the temperature range of 280 to 350 °C, also observed for the xylitol sample (Figure 6a). Similarities are also shown in these samples’ Ozawa–Flynn–Wall analyses, with their isoconversions showing greater parallelism between 220 and 350 °C. This suggests the predominance of xylitol decomposition in the thermal decomposition profile of the E4 sample. Table 4 shows that the E4 sample presented slightly higher E_a_ values than the pure xylitol sample (C_5_H_12_O_5_); it requires more energy to initiate the decomposition process.

This suggests that despite having a high xylitol content in its composition, the steviol glycosides influence the decomposition process of the E4 sample. The thermal profile of the curves observed by the Friedman method shows similarities between the samples (stevioside and E3) since the elongation in the central degradation region differs from what was observed for the samples with polyols (E2 and E4). The influence of the added compound present in the tabletop sweetener (erythritol or xylitol) facilitated the decomposition process, which the reduced E_a_ and ln A values can confirm.

According to the values observed for the activation energies for all the samples studied (as shown in Table 2, Table 3 and Table 4 and Appendix A), the Friedman method exhibited the most significant variation in E_a_. This suggests that the Ozawa–Flynn–Wall method better fits the sweetener samples’ thermal decomposition profile.

According to relevant studies in thermal decomposition kinetics through thermal analysis [31,32,33,34] the use of nonlinear regression for a series of measurements with different heating ratios or different reaction temperatures (multivariate nonlinear regression) allows the selection of the most straightforward and most appropriate formal kinetic model and a reliable estimate of the activation parameters. Through this method, the entire reaction is described by a combination of formal reaction steps (independent, parallel, competitive, or consecutive) with constant activation parameters, varying the type of combination of the steps described. However, it is essential to emphasize that these steps may not be interpreted as existing chemical reactions. A value considered excellent for the goodness-of-fit for the best model does not necessarily mean that the kinetic description is correct from a physicochemical point of view [35,36]. Kinetic modeling using the multivariate nonlinear regression method was performed using the Kinetics Neo software. The primary purpose of using this resource is not to determine the actual model that describes the material decomposition process but rather to determine a model that describes this process in the best possible way, thus allowing the prediction of the course of the reaction.

The Model-based package was used to investigate the reaction mechanism of the decomposition of tabletop sweetener samples. The kinetic models tested were consecutive, parallel, and competitive steps for non-autocatalytic reactions (Fn) and autocatalytic reactions (Cn). Figure 7, Figure 8, Figure 9, Figure 10, Figure 11 and Figure 12 exhibit the selected models for reactions that demonstrated the best fit after extensive simulations for each one of the sweeteners based on steviol glycosides.

To determine the optimal fit for the decomposition mechanism of each sample, simulations were conducted in which the reaction order, the number of decomposition steps, and the type of reaction were varied. The best model for each sample was identified, enabling the suggestion of the decomposition mechanism based on the correlation coefficient R^2^. It is worth mentioning that the relevance of the model was evaluated using the F-test statistics provided by the software.

The best fit for each sweetener is highlighted in Figure 7, Figure 8, Figure 9, Figure 10, Figure 11 and Figure 12, along with the calculated R^2^ values. The three best kinetic models identified for each sweetener sample are presented in Table 5 and Table 6, but not in Table 7. It is worth noting that for the xylitol sample (Table 7), no more than two models were found whose R^2^ and F-test were adequate. The selection of the best kinetic model was based on the highest calculated R^2^ value, which was further validated by the F-test statistics.

Simulation of the kinetics model for the stevioside sample occurs in two consecutive stages using both reaction order Fn and one competitive step, Fn (Figure 7). Figure 8 shows the simulation result for the kinetic model of sample E3. The most appropriate kinetic model to describe the thermal decomposition reaction of the E3 sample was the one with three consecutive reaction steps: the first and third ones of n order and the second stage, an autocatalytic reaction. On the other hand, the most appropriate kinetic model to describe the decomposition reaction of the E2 sample (Figure 9) was two consecutive stages, (Fn) followed by a (Cn). Erythritol also presented the same reaction order as this last one. Sample E4 had its thermal decomposition reaction described by a kinetic model (Figure 11) of three consecutive stages (FnFnFn). Xylitol (Figure 12) presented the best fit in two competitive steps (FnFn).

Investigating the reaction models of the sweetener samples containing erythritol, stages were observed in which autocatalytic reactions occurred in the samples E2 and erythritol. The formation of ethylene glycol, acetone, or formaldehyde can explain this. The latter two, in turn, can act as catalysts for secondary reactions (reduction or generation of free radicals) in the erythritol decomposition process. The sample containing stevia E3 exhibits an autocatalytic reaction in one stage. During its decomposition, ketones, aldehydes, and acetic and formic acids can be formed.

As the kinetic study by thermal analysis aimed to evaluate the thermal stability and decomposition process of the materials, it was possible to verify that both stevia and erythritol, separately, have autocatalytic stages, which can produce undesirable compounds or unpleasant flavors when subjected to high temperatures during the preparation of food and beverages. Thus, it is understood that one of the most suitable alternatives to mitigate such effects would be the combination of stevia with polyol, confirmed by the lower values of the autocatalytic reaction order. It can be identified that there is a synergistic effect between stevia and erythritol. Erythritol increases the thermal stability of stevia, and this, due to the presence of active compounds (stevioside and rebaudioside), can act as antioxidants, preventing the release of undesirable volatile compounds [37,38].

## 4. Conclusions

Kinetic parameters, such as activation energy and pre-exponential factor are key to determining solid-phase reaction mechanisms. The kinetic data of a material in a solid state are of primary interest in many technological processes, as they can help decide on its useful life, understand its thermal stability, and assist in quality control in the industry. In the case of the present paper, which investigated the thermal decomposition of some tabletop sweeteners based on steviol glycosides through isoconversional methods, it was possible to determine how combinations with stevia influence the decomposition process—the reaction rate, reaction order, and products formed.

The kinetic evaluation shows an interesting conclusion that the combination of stevia with polyol will be the most favorable for food submitted to temperature. These two classes of sweeteners in the same final product prevent the production of undesirable compounds or unpleasant flavors in contrast if they were alone. This brings crucial information to the food industry during the production and processing of natural sweeteners.

## Figures and Tables

**Figure 1 foods-14-01233-f001:**
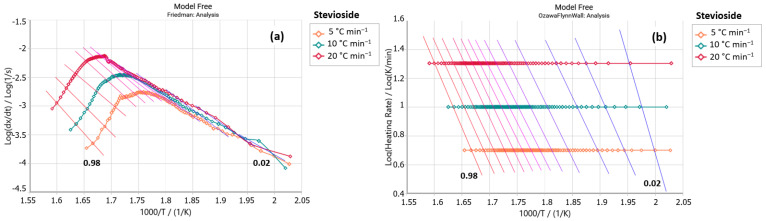
(**a**) Friedman and (**b**) Ozawa–Flynn–Wall analysis for stevioside (*y* axis: reaction rate; *x* axis: Temperature).

**Figure 2 foods-14-01233-f002:**
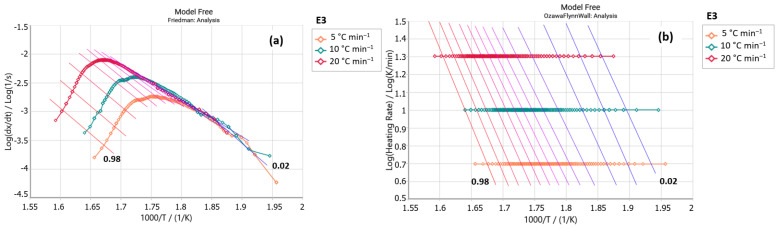
(**a**) Friedman and (**b**) Ozawa–Flynn–Wall analysis for sample E3 (steviol glycosides (*y* axis: reaction rate; *x* axis: Temperature).

**Figure 3 foods-14-01233-f003:**
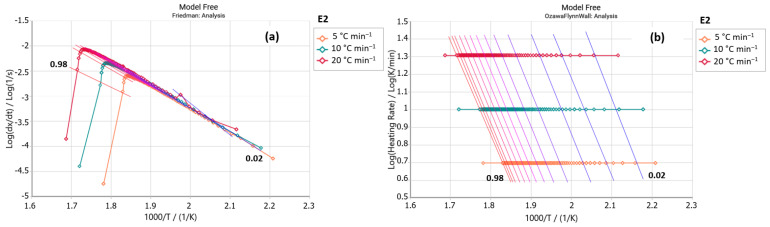
(**a**) Friedman and (**b**) Ozawa–Flynn–Wall analysis for sample E2 (steviol glycosides + erythritol) (*y* axis: reaction rate; *x* axis: Temperature).

**Figure 4 foods-14-01233-f004:**
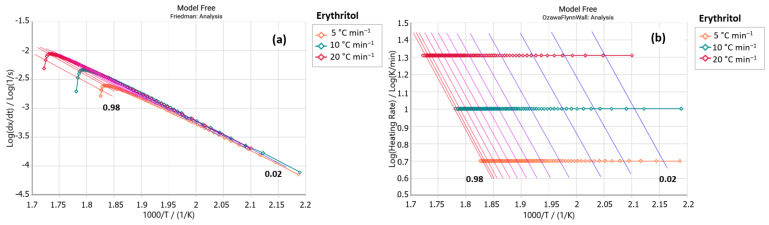
(**a**) Friedman and (**b**) Ozawa–Flynn–Wall analysis for sample erythritol (*y* axis: reaction rate; *x* axis: Temperature).

**Figure 5 foods-14-01233-f005:**
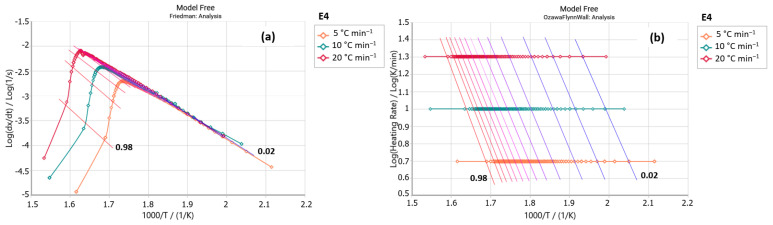
(**a**) Friedman and (**b**) Ozawa–Flynn–Wall analysis for sample E4 (steviol glycosides + xylitol) (*y* axis: reaction rate; *x* axis: Temperature).

**Figure 6 foods-14-01233-f006:**
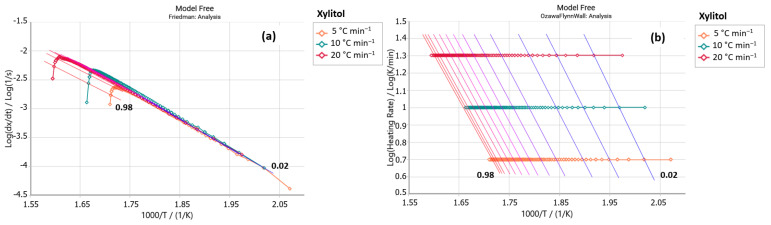
(**a**) Friedman and (**b**) Ozawa–Flynn–Wall analysis for xylitol sample (*y* axis: reaction rate; *x* axis: Temperature).

**Figure 7 foods-14-01233-f007:**
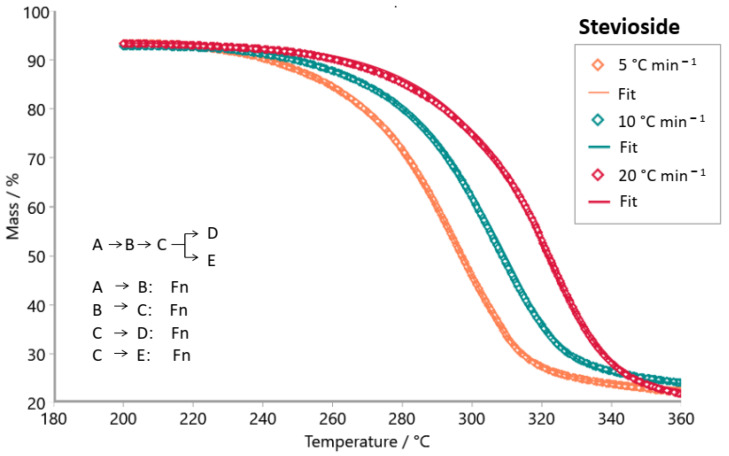
Simulation of kinetics model for stevioside sample in two consecutive steps using Fn (1st stage), Fn (2nd stage), and one competitive using Fn (3rd stage), Fn (4th stage); R^2^ = 0.99986. (**◊** Experimental data; **―** best fit after simulation).

**Figure 8 foods-14-01233-f008:**
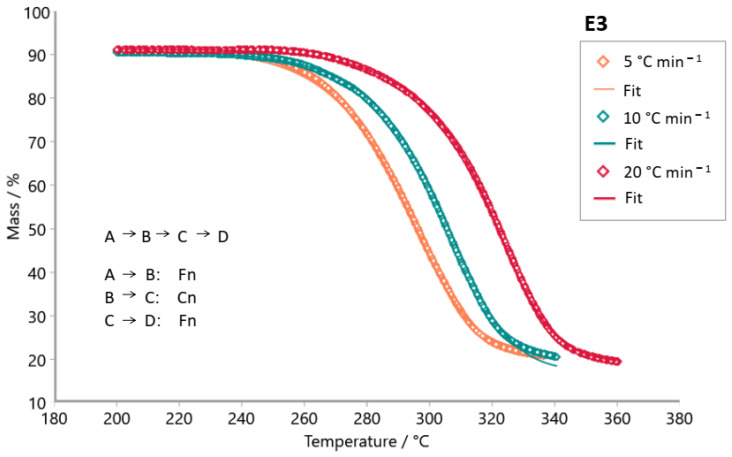
Simulation of kinetics model for sample E3 (steviol glycosides) in three consecutive steps using Fn (1st stage), Cn (2nd stage), and Fn (3rd stage); R^2^ = 0.99965. (**◊** Experimental data; **―** best fit after simulation).

**Figure 9 foods-14-01233-f009:**
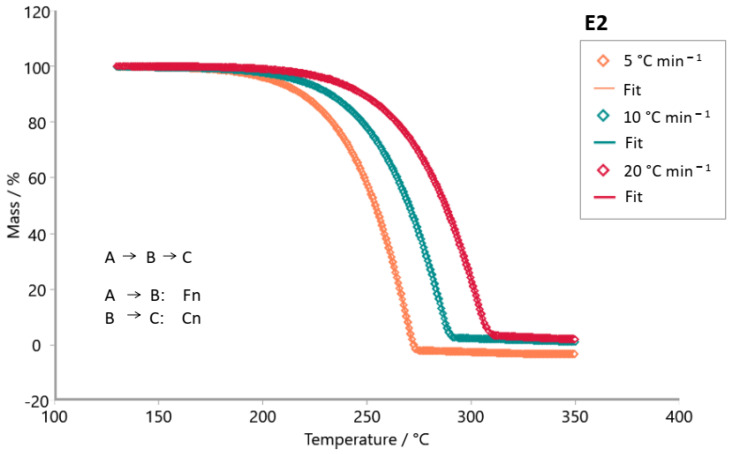
Simulation of kinetics model for sample E2 (steviol glycosides + erythritol) in two consecutive steps using Fn (1st stage) and Cn (2nd stage), R^2^ = 0.99997. (**◊** Experimental data; **―** best fit after simulation).

**Figure 10 foods-14-01233-f010:**
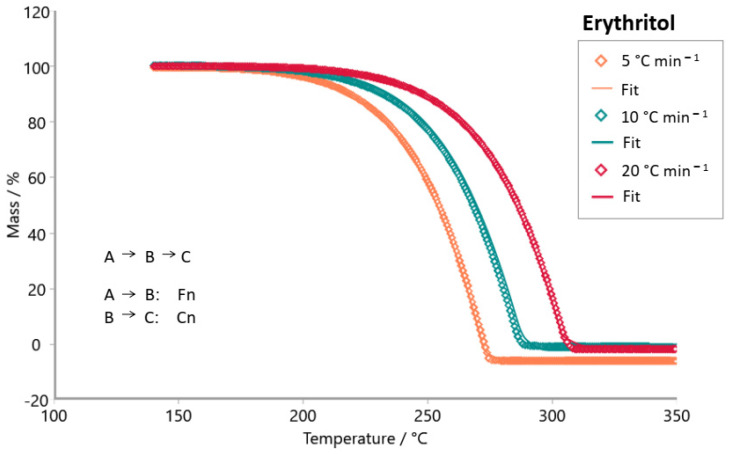
Simulation of kinetics model for erythritol sample in two consecutive stages using Fn (1st stage), and Cn (2nd stage), R^2^ = 0.99966. (**◊** Experimental data; **―** best fit after simulation).

**Figure 11 foods-14-01233-f011:**
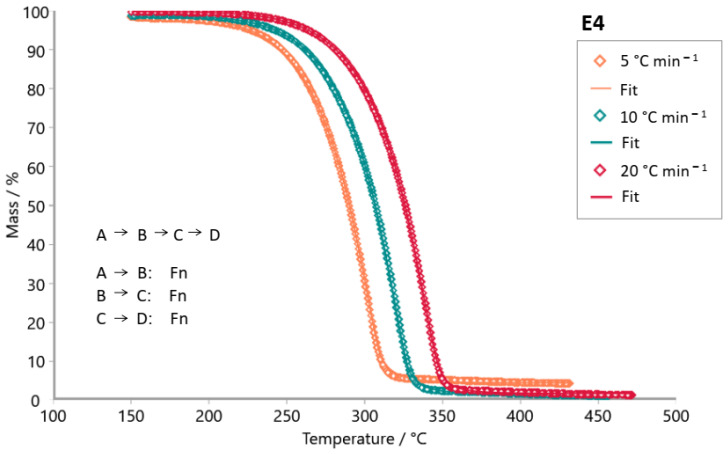
Simulation of kinetics model for sample E4 (steviol glycosides + xylitol) in three consecutive steps using Fn (1st stage), Fn (2nd stage), and Fn (3rd stage), R^2^ = 0.99997. (**◊** Experimental data; **―** best fit after simulation).

**Figure 12 foods-14-01233-f012:**
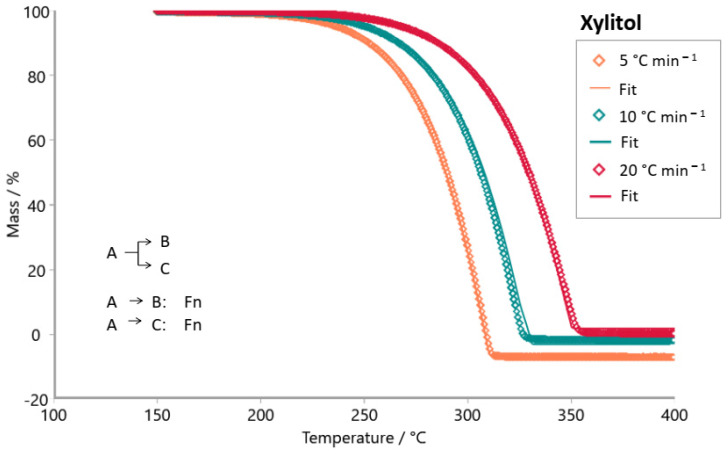
Simulation of kinetics model for xylitol sample in two competitive steps using Fn (1st stage) and Fn (2nd stage), R^2^ = 0.99957. (**◊** Experimental data; **―** best fit after simulation).

**Table 1 foods-14-01233-t001:** Some properties and characteristics of natural sweeteners [17,18,19,20,21,22].

Sweetener	Chemical Structure	Characteristics
steviol glycosides	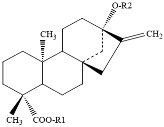	▪ 200 to 450 times the sweetness of sucrose; ▪ Stevioside and Rebaudioside A are the glycosides present in the highest content (≥95%) in *Stevia* extract; ▪ Bitter taste; ▪ Acid/alkaline stability: 4–6.
erythritol	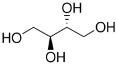	▪ 60–70% as sweet as sucrose; ▪ No aftertaste, odorless; ▪ Sugar-like texture; ▪ Acid/alkaline stability: 2–10.
xylitol	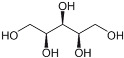	▪ Similar to sucrose (100% sweetness); ▪ Odorless; ▪ Acid/alkaline stability: 2–10; ▪ Sugar-like texture.

**Table 2 foods-14-01233-t002:** Activation energy and pre-exponential factor obtained by Friedman and OFW methods for stevioside and E3 (steviol glycosides) samples.

	Friedman				Ozawa-Flynn-Wall			
	Stevioside		E3		Stevioside		E3	
α	E_a_ [kJ mol^−1^]	ln A [s^−1^]	E_a_ [kJ mol^−1^]	ln A [s^−1^]	E_a_ [kJ mol^−1^]	ln A [s^−1^]	E_a_ [kJ mol^−1^]	ln A [s^−1^]
0.1	151.65	11.62	128.43	9.30	146.37	11.18	144.23	10.73
0.2	144.79	10.89	124.47	8.99	145.27	10.99	134.51	9.84
0.3	130.58	9.56	130.94	9.62	142.16	10.66	132.21	9.66
0.4	136.83	10.17	142.68	10.73	139.24	10.38	133.18	9.78
0.5	144.88	10.93	151.71	11.57	140.18	10.47	136.31	10.09
0.6	163.87	12.66	159.40	12.27	145.90	11.00	140.22	10.46
0.7	174.36	13.57	177.02	13.83	152.24	11.58	145.92	10.99
0.8	178.24	13.87	178.24	13.92	159.58	12.23	152.81	11.62
0.9	224.30	17.70	229.58	18.27	170.93	13.19	167.94	12.95

**Table 3 foods-14-01233-t003:** Activation energy and pre-exponential factor were obtained by Friedman and OFW methods for E2 (steviol glycosides + erythritol) and erythritol samples.

	Friedman				Ozawa-Flynn-Wall			
	E2		Erythritol		E2		Erythritol	
α	E_a_ [kJ mol^−1^]	ln A [s^−1^]	E_a_ [kJ mol^−1^]	ln A [s^−1^]	E_a_ [kJ mol^−1^]	ln A [s^−1^]	E_a_ [kJ mol^−1^]	ln A [s^−1^]
0.1	97.67	7.05	97.41	6.98	100.27	7.34	101.26	7.44
0.2	96.24	6.89	97.78	7.04	99.41	7.27	101.42	7.46
0.3	96.53	6.96	97.66	7.07	98.80	7.22	101.69	7.51
0.4	94.51	6.82	98.45	7.19	98.46	7.21	102.18	7.58
0.5	93.36	6.77	99.01	7.31	98.08	7.21	102.73	7.66
0.6	92.85	6.81	99.74	7.46	97.70	7.21	103.36	7.75
0.7	91.83	6.82	100.49	7.64	97.33	7.22	104.03	7.86
0.8	92.26	7.01	101.38	7.88	96.98	7.24	104.74	7.99
0.9	91.96	7.18	102.88	8.28	96.51	7.29	105.52	8.15

**Table 4 foods-14-01233-t004:** Activation energy and pre-exponential factor obtained by Friedman and OFW methods for sample E4 (steviol glycosides + xylitol) and xylitol.

	Friedman				Ozawa-Flynn-Wall			
	E4		Xylitol		E4		Xylitol	
α	E_a_ [kJ mol^−1^]	ln A [s^−1^]	E_a_ [kJ mol^−1^]	ln A [s^−1^]	E_a_ [kJ mol^−1^]	ln A [s^−1^]	E_a_ [kJ mol^−1^]	ln A [s^−1^]
0.1	97.22	6.34	95.21	6.07	96.92	6.37	98.86	6.48
0.2	95.83	6.23	92.48	5.87	96.77	6.39	96.59	6.31
0.3	95.62	6.24	90.91	5.78	96.54	6.39	95.34	6.23
0.4	97.22	6.43	89.71	5.74	96.79	6.44	94.42	6.18
0.5	99.87	6.74	88.66	5.72	97.49	6.53	93.70	6.16
0.6	102.78	7.08	87.87	5.75	98.47	6.65	93.08	6.15
0.7	104.52	7.33	86.99	5.79	99.59	6.80	92.52	6.16
0.8	107.90	7.77	85.67	5.84	100.25	6.92	91.90	6.18
0.9	117.41	8.75	87.51	6.27	102.29	7.15	91.19	6.21

**Table 5 foods-14-01233-t005:** Kinetic model tested for a stevioside sample to reactions combined in consecutive followed by a competitive stage.

Sample	Kinetic Model	R^2^	F_exp_	E_a_/[kJ mol^−1^]				ln A/[s^−1^]				Reaction Order			
				E_a1_	E_a2_	E_a3_	E_a4_	A_1_	A_2_	A_3_	A_3_	n_1_	n_2_	n_3_	n_4_
stevioside	F1Fn	0.99983	1.206	123.66	193.85	-	-	9.27	15.64	-	-	1	1.80	-	-
	FnFnFn	0.99986	1.017	123.08	133.51	170.15	-	9.52	10.19	13.37	-	1.38	0.47	1.40	-
	FnFnFnFn	0.99986	1.000	118.34	209.53	319.72	69.08	8.73	17.27	27.64	4.07	0.94	1.19	5.71	1.41

**Table 6 foods-14-01233-t006:** Kinetic model tested for tabletop sweetener samples to reactions in consecutive stages.

Sample	Kinetic Model	R^2^	F_exp_	E_a_/[kJ mol^−1^]			ln A/[s^−1^]			Reaction Order		
				E_a1_	E_a2_	E_a3_	A_1_	A_2_	A_3_	n_1_	n_2_	n_3_
E2	F_1_Fn	0.99920	23.91	102.85	110.52	-	7.82	8.30	-	1	0.30	-
	FnFn	0.99985	4.49	84.56	120.54	-	5.84	9.33	-	0.39	0.44	-
	FnCn	0.99997	1.00	95.68	94.54	-	6.96	6.43	-	0.57	0.01	-
E3	F1Fn	0.99940	1.72	143.04	139.80	-	10.99	10.80	-	1	1.55	-
	FnCn	0.99942	1.68	137.55	133.32	-	10.49	10.15	-	0.97	1.62	-
	FnCnFn	0.99965	1.00	141.05	55.88	162.20	10.87	2.81	12.64	0.91	0.36	1.36
E4	F1Fn	0.99962	12.37	100.82	113.65	-	6.95	7.87	-	1	0.32	-
	FnFn	0.99992	2.44	86.11	122.60	-	5.45	8.71	-	0.42	0.50	-
	FnFnFn	0.99997	1.00	88.80	103.48	147.41	5.72	6.92	11.68	0.50	0.14	1.86
erythritol	F1Fn	0.99894	3.09	104.63	116.87	-	7.99	8.89	-	1	0.28	-
	FnFn	0.99955	1.34	99.79	−175.54	-	7.22	−17.08	-	0.27	2.70	-
	FnCn	0.99966	1.00	98.56	100.16	-	7.26	6.98	-	0.2	0.01	-

**Table 7 foods-14-01233-t007:** Kinetic model tested for tabletop sweetener samples to reactions in competitive stages.

Sample	Kinetic Model	R^2^	F_exp_	E_a_/[kJ mol^−1^]			ln A/[s^−1^]			Reaction Order		
				E_a1_	E_a2_	E_a3_	A_1_	A_2_	A_3_	n_1_	n_2_	n_3_
xylitol	F1Fn	0.99957	1.000	85.77	94.42	-	−10.41	5.96	-	1	0.08	-
	FnFn	0.99957	1.002	96.88	93.97	-	5.15	5.88	-	0.25	0.06	-

## Data Availability

The original contributions presented in the study are included in the article/Appendix A, further inquiries can be directed to the corresponding author.

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
