# Peer review of "Investigation and Determination of Kinetic Parameters of Sweeteners Based on Steviol Glycosides by Isoconversional Methods"

_foods, 2025, doi:10.3390/foods14071233_

Round 1
Reviewer 1 Report
Comments and Suggestions for Authors
Manuscript Number: foods-3505241
Title: Investigation and determination of kinetic parameters by isoconversional methods of sweeteners based on Steviol Glycosides
Journal: foods
Reviewer’s Comments :
Dear Editor,
Dear Authors,
Manuscript entitled “Investigation and determination of kinetic parameters by isoconversional methods of sweeteners based on Steviol Glycosides” was thoroughly reviewed. In this work, The decomposition processes of tabletop sweeteners based on steviol glycosides were investigated to determine the kinetic parameters: activation energy (Ea) and the logarithm of the pre-exponential factor (ln A) based on the converted fraction (a). These parameters were assessed using the Friedman and Ozawa–Flynn–Wall isoconversion methods with the NETZSCH Kinetics Neo software and the Model Free package.
Highlights and features of this work are as follows: “The results indicated that both stevioside and steviol glycosides (E3) samples require higher energy to initiate their decomposition. Furthermore, the samples based on steviol glycosides exhibited distinct probable decomposition mechanisms.”
- As is well known, the iterative of Ozawa-Flynn-Wall (OFWI) method yields more precise results than the Ozawa-Flynn-Wall (OFWI) method when calculating activation energy. So it is suggested that the authors should try to use the OFWI method to solve the activation energy.
- As is well known, the Masterplots method can easily obtain accurate mathematical equations for the mechanism function. So it is suggested that the authors should try to use the Masterplots method to solve the mechanism equation..
- The α-Eα curve can provide important information. Please ask the author to supplement the curves corresponding to the two methods
Reviewer
Feb. 20, 2025.
Author Response
We thank the reviewers for their constructive suggestions, which we have carefully considered in our revisions. We are submitting the revised version of our manuscript for the journal's consideration, with all new sentences highlighted in yellow.
Comment 1. As is well known, the iterative of Ozawa-Flynn-Wall (OFWI) method yields more precise results than the Ozawa-Flynn-Wall (OFWI) method when calculating activation energy. So it is suggested that the authors should try to use the OFWI method to solve the activation energy.
Response: We greatly appreciate the feedback and agree with the suggestions made. The authors already used the interactive OFW method in Kinetics Neo software. However, we forgot to include the term “interactive" in the main text. Therefore, we added a note on page 5 to clarify that the method is interactive.
Comment 2. As is well known, the Masterplots method can easily obtain accurate mathematical equations for the mechanism function. So it is suggested that the authors should try to use the Masterplots method to solve the mechanism equation.
Response: We also appreciate the recommendation of the reviewer. The master plot corresponds to the shape of reaction type only for the multi-point differential model-free analysis methods (i.e. Friedman), and it can not provide reliable information for the integral model-free methods (i.e. Ozawa-Flynn-Wall). So we decided to not insert the master plots into the manuscript because OFW method was the best fit for the sweteners samples in the present study.
Comment 3. The α-Ea curve can provide important information. Please ask the author to supplement the curves corresponding to the two methods.
Response: The α-Ea curves for each sample are now included in Supplementary Material S1 (Figure S1 to S6) and are referenced in the text on page 8.
Reviewer 2 Report
Comments and Suggestions for Authors
Dear authors,
The present paper deals with the determination of the kinetic parameters (activation energy and pre-exponential factor) of six different samples of natural-origin tabletop sweeteners based on steviol glycosides using Kinetics Neo soft-ware.
The paper presents an interesting subject covering the aspects of thermal decomposition of the samples in three different heating rates. Although, authors should explain the following (for details, please see the comments inside the pdf text that is uploaded with the present comments):
-Why do you use only these two models compared to others not only model dependent but also non – dependent models?
- Many figures and tables are not explained well or not explained at all in the text (see tables 5,6,7).
-Authors do not present the initial experimental TG and DTG curves, which are very important for the analysis of the study?
-Please explain the symbols and correct the mathematical expressions that are presented in the Material and Method section
-Is it sufficient to use only three heating rates for this analysis? Please explain that with the addition of references that support this opinion.

Author Response
We thank the reviewers for their constructive suggestions, which we have carefully considered in our revisions. We are submitting the revised version of our manuscript for the journal's consideration, with all new sentences highlighted in yellow.
Comment 1. Why do you use only these two models compared to others not only model dependent but also non – dependent models?
Response: We sincerely appreciate the reviewer's insightful comments. The authors chose the non-isoconversional methods by Friedman and Ozawa-Flynn-Wall because they are suitable for corroborating information about samples that undergo complex multi-step decomposition processes.
Comment 2. Many figures and tables are not explained well or not explained at all in the text (see tables 5,6,7).
Response: We are grateful for the feedback and agree with the suggestion made. Thus, we have enriched the manuscript by incorporating a detailed discussion of Tables 4, 5, and 6 on page 8.
Comment 3. Authors do not present the initial experimental TG and DTG curves, which are very important for the analysis of the study?
Response: We completely agree with the reviewer regarding the critical role of TG/DTG results in our kinetic study. As such, the relevant TG/DTG curves were included in a prior published study as described below:
Santana, N. da S.; Mothé, C.G.; de Souza, M.N.; Mothé, M.G. An Investigation by Thermal Analysis of Glycosidic Natural Sweeteners. J Therm Anal Calorim 2022, 147, 13275–13287, doi:10.1007/s10973-022-11550-x.
Comment 4. Please explain the symbols and correct the mathematical expressions that are presented in the Material and Method section
Response: We have meticulously included all symbols in the mathematical equations (Eq. 1 to 13) in the manuscript.
Comment 5. Is it sufficient to use only three heating rates for this analysis? Please explain that with the addition of references that support this opinion.
Response: As established in previous literature, utilizing at least three heating ratios is crucial for ensuring the reliability of results obtained by the Friedman and Ozawa-Flynn-Wall models when determining kinetic parameters. The authors followed carefully the methods described in the prestigious literature below:
Opfermann, J.; Kaisersberger, E.; Flammersheim, H.J. Model-free analysis of thermoanalytical data-advantages and limitations. Thermochim Acta 2002, 1-2, 12, 119-127, 2002, doi:10.1016/S0040-6031(02)00169-7
Vyazovkin, S.; Burnham, A.K.; Favergeon, L.; Koga, N.; Moukhina, E.; Pérez-Maqueda, L.A.; Sbirrazzuoli, N. ICTAC Kinetics Committee Recommendations for Analysis of Multi-Step Kinetics. Thermochim Acta 2020, 689, 178597, doi:10.1016/j.tca.2020.178597.
